# Comparative Analysis of Stk11/Lkb1 versus Pten Deficiency in Lung Adenocarcinoma Induced by CRISPR/Cas9

**DOI:** 10.3390/cancers13050974

**Published:** 2021-02-26

**Authors:** Martin F. Berthelsen, Siv L. Leknes, Maria Riedel, Mette A. Pedersen, Justin V. Joseph, Henrik Hager, Mikkel H. Vendelbo, Martin K. Thomsen

**Affiliations:** 1Department of Clinical Medicine, Aarhus University, DK-8200 Aarhus N, Denmark; mfb@biomed.au.dk (M.F.B.); maria.riedel@biomed.au.dk (M.R.); jvj@clin.au.dk (J.V.J.); 2Department of Biomedicine, Aarhus University, DK-8000 Aarhus, Denmark; siv.lund.leknes@post.au.dk (S.L.L.); meabpe@biomed.au.dk (M.A.P.); mhve@biomed.au.dk (M.H.V.); 3Department of Nuclear Medicine & PET Centre, Aarhus University Hospital, DK-8200 Aarhus N, Denmark; 4Steno Diabetes Center Aarhus, DK-8200 Aarhus N, Denmark; 5Department of Clinical Pathology, Vejle Hospital, Beriderbakken 4, DK-7100 Vejle, Denmark; Henrik.Hager@rsyd.dk; 6Aarhus Institute of Advanced Studies (AIAS), Aarhus University, DK-8000 Aarhus, Denmark

**Keywords:** lung cancer, CRISPR, adenocarcinoma, mouse model, STK11, PTEN

## Abstract

**Simple Summary:**

Lung cancer is by far the leading cause of cancer induced mortality worldwide with a median five-year survival rate of 19 percent. Genome sequencing of lung cancer samples has revealed several key mutated genes, which could be implicated in lung cancer formation. This study applied a mouse model of lung cancer based on CRISPR/Cas9 technology to functionally address key regulators of the mTor pathway, STK11 and PTEN. Our study revealed that loss of Stk11 drives lung adenocarcinoma progression, whereas Pten is dispensable. These functional mouse studies reveal that loss of Pten is non-essential for lung adenocarcinoma, which is in agreement with the low mutation rates of PTEN in human adenocarcinoma. In contrast, loss of Stk11 drive tumor progression and is often found mutated in human samples of lung adenocarcinoma.

**Abstract:**

This study focused on STK11, PTEN, KRAS, and TP53, which are often found to be mutated in lung cancer. We compared Stk11 and Pten implication in lung cancer in combination with loss of Trp53 and gain of function of Kras in a CRISPR/Cas9 mouse model. Mice with loss of Stk11, Trp53, and KrasG12D mutation (SKT) reached human endpoint at around four months post-initiation. In comparison, mice with loss of Pten, Trp53, and KrasG12D mutation (PKT) survived six months or longer post-initiation. Pathological examination revealed an increase in proliferation in SKT deficient lung epithelia compared to PKT. This difference was independent of Pten loss, indicating that loss of Pten is dispensable for cell proliferation in lung adenocarcinoma. Furthermore, tumors with loss of Stk11, Trp53, and KrasG12D mutation had a significantly higher progression rate, monitored by PET/MRI scanning, compared to mice with loss of Pten, Trp53, and KrasG12D mutation, revealing that mutations in Stk11 are essential for adenocarcinoma progression. Overall, by using the CRISPR/Cas9 mouse model of lung adenocarcinoma, we showed that mutations in Stk11 are a key driver, whereas loss of Pten is dispensable for adenocarcinoma progression.

## 1. Introduction

Although detection and treatment of lung cancer have significantly improved over recent decades, this malignancy is responsible for most cancer related mortality worldwide [1]. The median five-year survival rate of 19 percent still leaves room for improvement and stresses the requirement to expand our understanding of the molecular mechanisms underlying the disease [2].

A variety of key mutated genes have been associated with subtypes of lung cancer. Among these are the tumor suppressor genes *PTEN* and *STK11/LKB1*. Both genes are directly involved in the negative regulation of the mTOR pathway, which induces cell transformation when constitutively activated. PTEN regulates the pathway by inhibiting the phosphorylation of AKT, which activates mTOR when phosphorylated [3]. *STK11* negatively regulates the mTOR pathway by activation of AMPK [4]. Mutations of *STK11* are more frequently found in lung cancer than loss of *PTEN*, indicating discrepancy in the molecular mechanism [5,6]. Furthermore, loss of *STK11* often co-occurs with gain-of-function mutations of *KRAS*, whereas loss of *PTEN* rarely occurs with activation of KRAS. *PTEN* loss is common in squamous cell cancers (SCC) and small cell lung cancer, whereas *STK11* is often found mutated in adenocarcinoma [5]. 

Genetically engineered mouse models (GEMM) have been crucial for the analysis of cancer related gene function in lung cancer [7]. Traditionally genetically modified mice with conditional alleles for *Stk11, Pten*, and *Kras^G12D^* haven been applied to study lung cancer initiation and progression [8,9]. These studies have confirmed the implication of these three factors in lung cancer initiation and that combination of multiple mutations accelerated cancer progression [10]. However, one drawback with these models is the lack of clonal selection associated with tumor progression, as the whole tumor has the same gene alteration. 

It has become possible to study tumor progression from a clonal selection perspective with the introduction of CRISPR/Cas9 technology. By delivery of multiple sgRNAs in single AAV particles to Cas9 transgenic mice, clones with different mutation profiles will be generated, as the CRISPR guide only induces mutations in a subset of the transduced cells [11]. This deficiency permits the study of crosstalk between essential genes in cancer formation. Hereby, the developed tumor harbors the optimal mutation profile for the given tumor type [12].

In this study, we compared Stk11 and Pten implication in lung cancer in combination with loss of Trp53 and gain of function of Kras in a CRISPR/Cas9 mouse model. Analysis of the Pan-Lung Cancer gene set by The Cancer Genome Atlas [5] indicates that *STK11* loss is positively correlated to *KRAS* mutations in adenocarcinoma, whereas loss of *PTEN* is associated with squamous cell cancers. Mice were generated by CRISPR with loss of Stk11, Trp53, and Kras^G12D^ mutation or loss of Pten, Trp53, and Kras^G12D^ mutation. Loss of Stk11 results in rapid tumor development where mice reach human endpoint four months after initiation. In comparison, loss of Pten in combination with Trp53 and Kras^G12D^ mutation delayed tumor growth when measured by PET/MRI scan and proliferation analysis. Furthermore, these mice reach earliest human endpoint six months after initiation. Analysis of CRISPR induced mutation and immunohistochemistry for Pten loss or p-Akt activation revealed that Pten was lost in approximately half of the samples. Interestingly, loss of Pten did not alter proliferation in the tumors, and the lung cancer progression was similar to tumor with only loss of Trp53 and a Kras^G12D^ mutation. Overall, the functional analysis reveals that loss of Stk11 in combination with loss of Trp53 and Kras activation promotes tumor growth. In contrast, loss of Pten is dispensable for lung adenocarcinoma development when mutated in combination with loss of Trp53 and Kras activation. These functional mouse studies interpret the mutation profile from the Pan-Lung Cancer cohort, where loss to STK11 is predominantly occurs in adenocarcinoma, whereas loss of Pten is a rare event in the combination of activated Kras mutation. 

## 2. Results

### 2.1. Differential Loss of STK11 and PTEN in Lung Cancer

In human lung cancer, *STK11, PTEN, TP53, and KRAS* are among the most frequently mutated genes. To address the possibility for co-occurrent mutations in these four genes, a data based analysis was performed. The Pan-Lung Cancer gene set by The Cancer Genome Atlas [5] was analyzed by cBioPortal. The data set revealed that *STK11, PTEN, TP53, and KRAS* were mutated in between nine and 68 percent of lung cancers, with *TP53* mutations having the highest incidence and *PTEN* the lowest in a cohort of 1144 lung cancer samples (of which 660 were adenocarcinomas and 484 SCC). Of these genes *TP53* was often found mutated in lung cancer regardless of subtype, showing that alteration in the P53 pathway is essential for lung cancer formation as seen for other cancer types [13]. Mutation analysis for *STK11* and *PTEN* shows a differential preference with loss of *STK11* predominately seen in adenocarcinoma and loss of *PTEN* in SCC (Figure 1). For *KRAS* the three known gain-of-function mutations in G12, G13, and Q61 [14] were selected and pooled together. Analysis for these gain-of-function mutations in *KRAS* showed concomitant *KRAS* mutations with loss of *STK11* in adenocarcinoma. In contrast, *KRAS* mutations in SCC did not co-occur with either *STK11* or *PTEN* mutations (Figure 1, Table 1). These analyses indicate that two key regulators of the mTOR pathway, STK11 and PTEN, have different functions in lung cancer biology and that *KRAS* and *PTEN* are mutually exclusive. 

### 2.2. Generation of Two AAV Vectors to Induce Lung Cancer

The difference of STK11 and PTEN mutations in lung cancer suggested a differential function of these key regulators of the mTOR pathway in lung cancer development. To address their implication in lung tumor formation, we applied multiplexed sgRNA delivery for lung tumor initiation. By applying mutations with CRISPR technology, multiple clones will be formed with different mutation profiles. This allow clones with the preferable mutational landscape to outcompete other clones and reveal unique mutation combination for lung cancer development and progression. Four different AAV vectors were generated to transduce the lung epithelia of LSL-Cas9 mice [15]. A control construct expressing YFP and two other constructs containing the following sgRNAs: *Stk11*, *Kras*, *Trp53,* and a *Kras^G12D^* repair template (SKT) and *Pten*, *Kras*, *Trp53*, and a *Kras^G12D^* repair template (PKT) were used. The SKT and PKT viral constructs also contained the Cre recombinase to activate Cas9 and EGFP expression in the lung epithelia of the transgenic mice (Figure 2A). The sgRNAs for *Stk11, Kras, Trp53*, and the introduction of Kras^G12D^ have been previously validated [15]. The sgRNAs against *Pten* were validated in mouse embryonic fibroblasts (MEFs), and TIDE analysis revealed an efficiency of 91 percent (Figure 2B). Hereby, two AAV constructs were generated to induce lung cancer by mutation of either *Stk11* or *Pten* in combination with loss of *Trp53* and alteration of *Kras*. 

### 2.3. Comparative Analysis of Tumor Progression between Loss of Stk11 and Pten

To induce lung cancer in Cas9 transgenic mice, AAV containing sgRNAs were administrated to the mice. Three weeks post-delivery, lung tissues were examined for the presence of AAV DNA and specific mutations generated by CRISPR. Viral DNA was present in the samples but CRISPR induced gene editing could not be detected at this time point (Appendix A). Mice were also analyzed by histology examination 10 weeks post-transduction with AAV. Here small lesions with altered cellular phenotype were observed in the lungs. These lesions were GFP positive, indicating that Cas9-EGFP expression had been activated in the transformed cells by AAV transduction (Figure 2C). These data confirm that AAV containing guide RNAs can transduce lung tissues and induce cellular transformation. 

To gain insight into *Stk11* and *Pten* implication in lung cancer formation, a direct comparison of the two viral constructs was performed. Tumor progression in the two groups was followed by ^18^F-FDG microPET/MRI scanning. Not all tumors were ^18^F-FDG avid due to small size; thus, tumor volume was quantified by MRI imaging (Figure 3A,B and Appendix A). Here, the size increased rapidly for SKT induced tumors. In contrast, tumor volumes in mice with loss of *Pten* (PKT) increased at a slower rate and did not reach a similar volume to SKT tumors (Figure 3B). The increased tumor volume in the *Stk11* deficient tumors also led to decreased survival. The median overall survival was four months for SKT induced tumors and six months for PKT tumor barring mice, whereas mice injected with AAV-YFP control particles did not develop lung tumors over a 6 month period (Figure 3C). Microscopic examination of the lung samples revealed few but large tumors when *Stk11* (SKT) was lost. In contrast, mutation of PKT induced multiple tumors, but these were in general small when compared to tumors from *Stk11* deficient samples (Figure 3D). This difference was observed even when samples were taken at human endpoint, which differed by two months between the genotypes. Pathological examination identified the tumors in both SKT and PKT induced mice, to be of lung adenocarcinoma, even though *Pten* is more associated with SSC [16]. A third group of mice with only loss of Trp53 and gain of function of Kras was generated (KT). Here lung tumors developed similarly to the PKT group with a similar survival and tumor size (Figure 3C and Appendix A). These analyses show that *Skt11* is an essential tumor suppressor to control lung adenocarcinoma, whereas the implication of *Pten* is minor. 

### 2.4. Loss of Stk11 Increases Proliferation in Lung Adenocarcinoma

To gain insights into the increase in tumor size observed with the loss of Stk11, tumor samples underwent molecular analysis. Tumor samples were analyzed for CRISPR/Cas9 introduced mutations to reveal the mutation profiles in the tumors that progressed. In the SKT tumors, 15 of 20 analyzed tumors had mutations in all three target genes. In the PKT samples, three out of six samples had mutations in all three target genes. Five of the 20 tumors from the SKT group displayed *Kras^G12D^* repair, and four of these samples were also mutated for *Stk11* and *Trp53*. In contrast, only one sample from the *Pten* construct (PKT) had a *Kras^G12D^* mutation, and here neither *Pten* nor *Trp53* was mutated (Figure 4A and Appendix A). This indicates the development of clones with different mutational landscape by the CRISPR mediated mutation. 

The sequencing data from the tumor biopsies showed CRISPR induced mutations. However, no mutation was detected in the target genes in some of the tumor samples, suggesting that normal lung tissues were analyzed. This was especially true for samples with loss of Pten, where tumor lesions were small. Therefore, immunohistochemical analysis was performed to validate activation of key pathways as a consequence of the mutated genes. Here an anti-Stk11 antibody stained negatively in *Stk11*^−/−^ tumors. The Erk pathway was activated in *KRAS^G12D^* positive tumors, and the Akt pathway was activated in *Pten* mutated tumors (Figure 4B and Appendix A). This analysis revealed that CRISPR induced mutations had occurred. Interestingly, approximately half of the PKT induced tumors did not stain positive for p-Akt, indicating the Pten mutations had only occurred in a subset of the tumors (Figure 4B). Increased tumor volume is often related to altered proliferation, and therefore samples were stained with the cell proliferation marker Ki67. Tumors negative for Stk11 had approximately a two-fold increase in Ki67 positive cells compared to tumors initiated with virus containing guides against *Pten*, *Trp53*, and *Kras*. Remarkably, analysis of the Ki67 positive cells did not differentiate with regard to p-Akt staining in PKT samples. Furthermore, the Ki67 positive cells in PKT samples were similar to tumors initiated with only *Trp53* and *Kras* mutations. This shows that loss of Pten does not affect proliferation capacity in this subtype of lung tumor (Figure 4C). Finally, Western blot analysis confirmed activated Erk in the tumor samples and increased p-Akt in some PKT samples (Figure 4D). Overall, these data show that loss of Stk11 in combination with alteration in Trp53 and Kras increased proliferation, whereas loss of Pten in the same setting is dispensable for cell division. 

The advanced tumor progression with loss of Stk11 resulted in a metastasis in the abdomen of one mouse. This metastasis was associated with the intestine of the mouse at the location of the kidney (Figure 5A). Sequencing of this metastatic lesion and the primary lung tumor revealed identical mutational profiles in the target genes of the three CRISPR guides (Figure 5B). These analyses clearly indicated that the origin of this metastasis was the lung.

## 3. Discussion

In this study, we made a direct comparison between the loss of *Stk11* and *Pten,* two key regulators of the mTOR pathway, in a *Trp53^−/−^* and K*ras^G12D^* setting in lung cancer development. By using CRISPR/Cas9 technology, we obtained a mixed pool of mutations in the targeted lung epithelial cells. In the STK group, the majority of the analyzed tumor samples had mutation in all three target genes, indicating that multiple mutations accelerate tumor progression as has been predicted for lung cancer [17]. We also identified tumors where not all of the three CRISPR guides induced mutations. The PCR based method to detect CRISPR induced mutations could result in overlooked mutations, as large deletions or chromosomal translocation will not be detected with this method. Likewise, naturally occurring mutations could have been introduced to CRISPR transformed cells and promoted tumor growth, as seen in other models of cancer. Activated KRAS is associated with lung cancer, and we observed that Kras has a gain-of-function mutation in approximately 25% of the samples. Gain-of-function mutations introduced by CRISPR occur in a low percentage of the cells due to the inefficient homology-directed repair [15]. Therefore, activation of Kras in a quarter of analyzed samples confirms Kras as an essential driver for lung cancer formation. Interestingly, our data indicated a preference for activation mutation of *Kras* to co-occur with loss of Stk11 as has been observed in more classical mouse studies [9]. In contrast, *Pten* mutations were not observed to be concomitant with K*ras^G12D^* mutation but were found to co-occur with loss of *Trp53*. Analysis of human lung cancer samples confirmed the preference for loss of *PTEN* together with *TP53* and co-occurrence of *KRAS* and *STK11* mutations. Loss of Trp53 and Kras was also present in the tumor samples but had a tendency to be mutually exclusive in patient samples. However, this was not always a strictly mutually exclusive event, as 10 percent of the human samples showed co-occurrence of *TP53, and KRAS* mutations (Table 1).

The clear difference in overall survival between loss of Stk11 versus Pten in combination with loss of Trp53 and Kras activation revealed that these m-TOR pathway regulators have distinct functions. This was also highlighted by the metastatic potential with loss of Stk11, which was not seen for loss of Pten. Loss of Pten is known to increase cell proliferation in other epithelial derived cancers, such as prostate cancer [3]. Our analysis showed a two-fold increase in proliferation with loss of Stk11 compared to loss of Pten. Interestingly, loss of Pten in the lung tumor did not increase proliferation, as the Ki67 positive cells were similar between tumors that were positive or negative for p-Akt, a marker for loss of Pten. However, Malkoski et al. showed that loss of Pten in the mouse lung epithelium can induce tumors after long latency, indicating a minor tumor suppressor function of Pten [18]. The non-essential role of Pten to drive tumor progression in combination with alteration in Trp53 and Kras can be associated with the tumor subtype or the lack of symbiosis with the other introduced mutations. As our model did not develop into SCC, where PTEN is often mutated, we could not address the function of PTEN in that subtype of lung cancer. Our data indicate that the function of Pten as a tumor suppressor gene in adenocarcinoma with alteration of Trp53 and Kras is minor. This is also in agreement with mutation analysis of different human lung cancer samples [6,16].

## 4. Materials and Methods 

### 4.1. Animals 

B6J.129(B6N)-Gt(ROSA)26Sortm1(CAG-cas9*,-EGFP)Fezh/J mice were purchased from Jackson Laboratories (catalog no. 26175).

### 4.2. sgRNA Design

The *Pten* sgRNA was designed using the (http://crispr.mit.edu) CRISPR design tool. See Appendix A for sgRNA sequence and genomic primers. The guide efficacy was determined with the Tracking of Indels by Decomposition (TIDE) software through transfection of LSL-Cas9 murine embryonic fibroblasts (MEFs) with pSpCas9(BB)-2A-GFP plasmid (Addgene ID: 48138) harboring the *Pten* sgRNA followed by FACS of GFP positive cells.

### 4.3. AAV Vector Constructs

The AAV:ITR-U6-sgRNA(Kras)-U6-sgRNA(p53)-U6-sgRNA(Lkb1)-pEFS-Rluc-2A-Cre-shortPA-KrasG12D-HDRdonor-ITR (SKT) viral vector was obtained from Addgene (ID: 60224). Replacing *Stk11* with *Pten* in the SKT viral vector generated the PKT construct. Removal of the Stk11 guide generated the Trp53-Kras gain-of-function construct. 

### 4.4. Cell Work

Primary MEFs were isolated from LSL-Cas9 mouse embryos. Single cells were obtained by dissection with scalpels and trypsin treatment. MEFs were grown in DMEM (Sigma-Aldrich, DK-2860, Søborg, Denmark) containing 1× penicillin–streptomycin (Sigma-Aldrich) and 10 percent FCS (Gibco, DK-4000, Roskilde, Denmark) at 37.5 °C. 

### 4.5. Fluorescence-Activated Cell Sorting

Cell sorting was carried out using a BD FACSAria III high speed cell sorter. GFP positive MEFs were analyzed at a velocity of 2000 cells/s, and 2000 cells were sorted into a culture plate well with DMEM.

### 4.6. AAV Production

Almost confluent HEK293T cells seeded on 15 cm dishes in 22 mL DMEM with 10 percent FCS and 1× penicillin–streptomycin were transfected with 14 µg Adeno-Helper plasmid, 14 µg serotype 9 capsid plasmid, 14 µg SKT or PKT plasmid, and 2.2 µg GFP plasmid (serving as transfection control) by using the branched MW 25,000 polyethylenimine (PEI) (Sigma-Aldrich, 408727) transfection reagent. The PEI mixture consisted of 350 µL PEI and 440 µL H2O. The DNA mixture consisted of 44 µg plasmid diluted in 760 µL H2O, and 790 µL 0.3 M NaCl was added to the PEI and DNA mixture just prior to mixing the two. Then, 3.1 mL of the resulting transfection mixture, which had been incubated for 10 min, was added to each dish. Cells were harvested two days post-transfection using a cell scraper and pelleted by centrifugation for 10 min at 400× *g*. The cell pellets were resuspended in PBS adjusted to 15 mL and subjected to five freeze–thaw cycles (−196 to 37 °C) for AAV release followed by centrifugation at 4000× *g* for five minutes. The supernatant encompassing the AAV particles was treated with benzonase (Sigma-Aldrich, DK-2860, Søborg, Denmark) for one hour at 37 °C at a concentration of 50 u/mL. Subsequently, the AAV sample was filtered through a 0.45 µm filter and then a 0.20 µm filter. Lastly, the AAV was concentrated by loading the sample on a 100K protein concentrator (Thermo Scientific, DK-4000, Roskilde, Denmark) by centrifugation at 4000× *g*. The AAV titer was evaluated by qPCR analysis of dilutions of the AAV sample and aligned to a plasmid standard curve. Brilliant III Ultra-Fast SYBR^®^ Green QPCR Master Mix reagent (Aqilent Technologies, DK-2600 Glostrup) was used according to the manufacturer’s recommendations. See Appendix A for primers.

### 4.7. Virus Delivery to the Lungs

Intranasal delivery of AAV was performed according to the previously established protocol [19]. In brief, three-month-old mice were anesthetized and set up in a biosafety cabinet. Titrated AAV solution was given in four times 5 µL sterile saline buffer and was pipetted directly into the nostrils of the mouse over a time course of 30 min. A titer of 10^11^ viral genomes was administered to each mouse. During the procedure, the mice were placed on a warm heating pad and afterwards under a heating lamp for recovery.

### 4.8. Histochemical Analysis

Tissue samples were fixed in four percent formaldehyde or paraformaldehyde (Sigma-Aldrich, DK-2860, Søborg, Denmark) overnight and embedded in paraffin before sections of four µm were cut. Antigen retrieval was performed at 100 °C in a citrate buffer at pH six for 20 min. Sections were blocked in 2.5 percent BSA (Sigma-Aldrich) in PBS with 0.1 percent Tween20 (Sigma-Aldrich, DK-2860, Søborg, Denmark) prior to probing with the following primary antibodies: GFP (CS-2956), Ki67 (MA5-14520), Lkb1 (SC-32245), phosphor-p44/42 MAPK (CS-9101), or p-Akt (CS-4060). Appropriate horseradish-peroxidase-conjugated secondary antibodies were used for development (Jackson ImmunoResearch, PA 19390, United States). Counterstaining was performed with hematoxylin and eosin or DAPI. 

### 4.9. Western Blot

Tissues were lysed in RIPA buffer containing phosphatase and protease inhibitors. Samples were sonicated, separated by gel electrophoresis, and transferred to a PVDF membrane before being blocked in 5% dry milk in TBS-T. Primary antibodies were applied overnight at 4 °C, and secondary antibodies were applied the following day before development. Primary antibodies: phosphor-p44/42 MAPK (CS-9101), p-Akt (CS-4060) and Vinculin (v9131, Sigma-Aldrich, DK-2860, Søborg, Denmark). 

### 4.10. PCR

PCR was performed on genomic DNA isolated from chorion villi lysis buffer treated MEFs or LSL-Cas9 mouse lung tumors/biopsies. Q5 High-Fidelity DNA Polymerase (NEB) was used according to the manufacturer’s recommendations. See Appendix A for primers.

### 4.11. ICE Analysis

Uploading Ab1 files from the Sanger sequencing of lung tumors to ICE v2 CRISPR Analysis Tool (https://synthego.com) identified mutations in the target genes.

### 4.12. MicroPET/MRI

The mice underwent functional positron emission tomography (PET) and 1T anatomical magnetic resonance imaging (MRI; Mediso Medical Imaging Systems). Anesthesia with isoflurane was initiated with the mouse placed in an acrylic glass chamber and maintained with respiration in a mask during the scan. A bolus of [^18^F]-Flurodeoxyglucose (FDG) (~15 MBq/animal) was injected via a tail vein catheter, and PET scanning was performed for 50 to 70 min after injection, followed by a 25 min T1 weighted MR-scan. Body temperature and respiration frequency were monitored during anesthesia.

A static PET image was reconstructed with a three-dimensional ordered subset expectation algorithm (Tera-Tomo 3D; Mediso Medical Imaging Systems) with four iterations and six subsets and a voxel size of 0.4 × 0.4 × 0.4 mm^3^. Data were corrected for dead-time, decay, and randomness using a delayed coincidence window without corrections for attenuation and scatter. A specialist in nuclear medicine identified tumors by visual inspection of the PET/MR-scans using Nucline v2.01 (Mediso Medical Imaging Systems). Borders of the tumors were drawn manually, and volume was calculated using HERMES (Hermes Medical Solutions).

### 4.13. Statistics

A log-rank test was used for statistical analysis of the Kaplan–Meier survival curve. An unpaired *t*-test was used for Ki67 quantification. A *p* value ≤ 0.05 was considered a statistically significant difference between the two groups. 

### 4.14. Study Approval

All animal experiments were conducted in accordance with the protocol approved by the Danish Animal Experiments Inspectorate (license no. 2017-15-0201-01244). Housing and care of the mice was in accordance with the Danish animal research proposal on genetically modified animals. Mice were euthanized by cervical dislocation.

## 5. Conclusions

In summary, we demonstrated that loss of Stk11 drives lung adenocarcinoma through increased proliferation in combination with loss of Trp53 and Kras activation. In contrast, loss of Pten is impaired for the development of lung adenocarcinoma when Trp53 and Kras are altered. By applying CRISPR/Cas9 technology to generate in vivo lung cancer models, it was possible to generate clones with different mutational profiles, which allows one to assess the implication of individual genes in cancer formation. The implication of STK11 mutations in lung adenocarcinoma in combination with gain of function of KRAS and loss of TP53 is common in human samples and confirms that CRISPR/Cas9 cancer models reflect human cancer biology. 

## Figures and Tables

**Figure 1 cancers-13-00974-f001:**
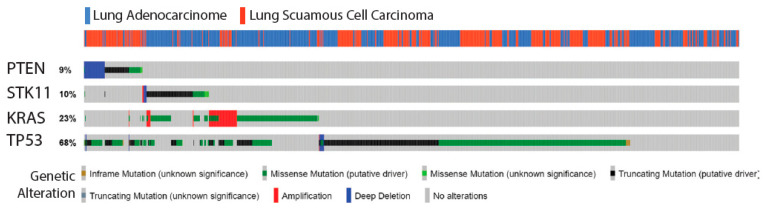
Mutation profile of *STK11*, *PTEN, KRAS*, and *TRP53* in human lung cancer. Genomic alterations of *STK11*, *PTEN*, *KRAS*, and *TP53* in The Pan-Lung Cancer gene set by The Cancer Genome Atlas (total *n* = 1184; adenocarcinoma = 660, squamous cell carcinoma = 484). Data were generated from cBioPortal.org.

**Figure 2 cancers-13-00974-f002:**
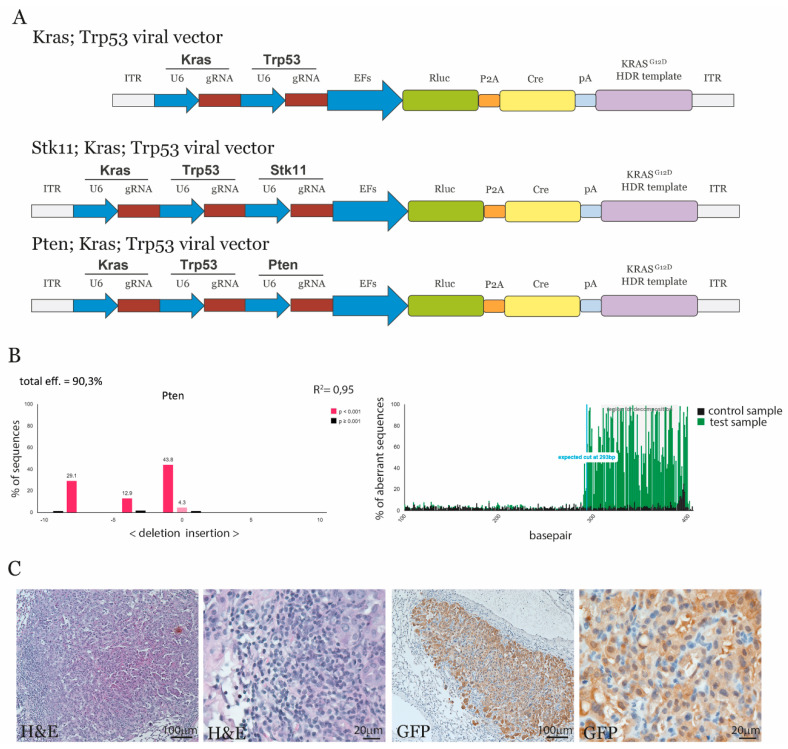
Induction of lung cancer by CRISPR/Cas9. (**A**) Three constructs for expression of sgRNAs and Cre by for AAV particles were generated. The first construct contains sgRNAs against *Kras* and *Trp53*; the second construct contains sgRNAs against *Stk11*, *Kras*, and *Trp53*; and a third construct contains sgRNAs against *Pten*, *Kras*, and *Trp53*. A U6 promoter expressed the sgRNAs, and an EF promoter expressed Cre expression. The construct also contains an 800 bp repair template for the induction of the *Kras^G12D^* mutation. (**B**) The efficiency of the Pten sgRNA was determined using the Tracking of Indels by Decomposition (TIDE) software for analysis. (**C**) Mice were inoculated with AAV particles containing the SKT viral construct, and lungs were examined 10 weeks post-treatment. Paraffin sections were stained with H&E and GFP antibody (brown stain) to confirm lung abnormalities and activation of Cas9/GFP expression in the transgenic mice (*n* = 5).

**Figure 3 cancers-13-00974-f003:**
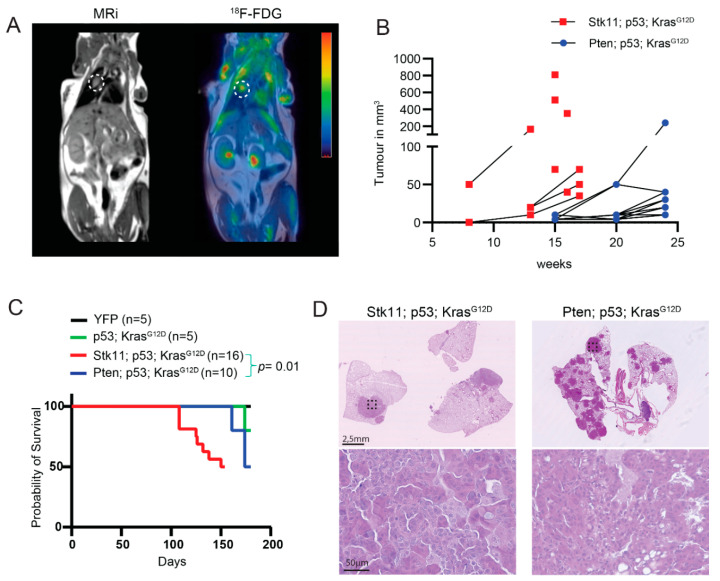
Loss of Stk11 accelerated lung tumor progression. Mice inoculated with AAV particles containing sgRNAs were followed for lung cancer progression by PET/MRI scanning. (**A**) Lung cancer was imaged by PET/MRI with ^18^F-FDG as tracer. The white dotted circle indicates the tumor. A representative picture is shown (*n* > 10). (**B**) Quantification of tumor volume by PET/MRI scanning (SKT *n* = 9, PKT *n* = 5). (**C**) Kaplan–Meier curve for overall survival for SKT (*n* = 16), PTK (*n* = 10), and KT (*n* = 5) lung cancer induced mice and control AAV-YFP treated mice (*n* = 5). (**D**) H&E stained paraffin section from lung samples at termination point. SKT samples were from 4 months and PTK samples from 6 months post-induction with viral particles. The dotted box marks the area of high magnification. Representative pictures are shown (*n* = 10).

**Figure 4 cancers-13-00974-f004:**
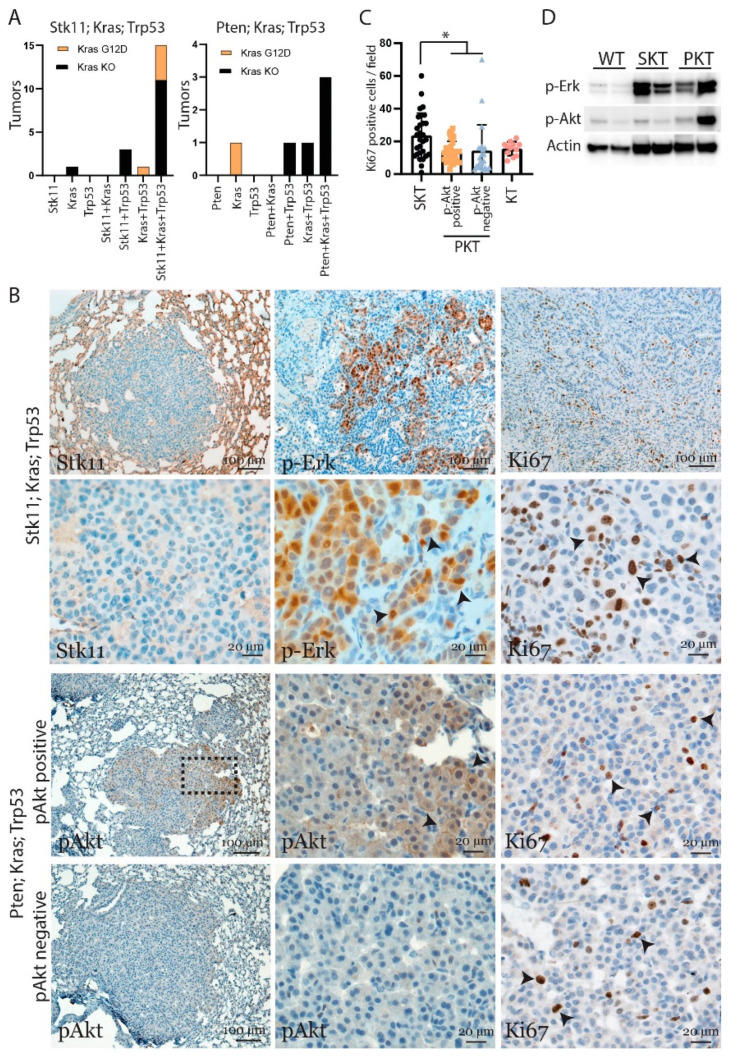
Loss of Stk11 increases proliferation to drive tumor progression. (**A**) Sanger sequencing was performed on tumor gain-of-function mutation. (**B**) Tissue sections of lung samples at termination point (SKT: 4 months, PKT: 6 months) were examined. Sections were stained with antibodies for Ki67, Stk11, p-AKT, and p-Erk (brown stain) and hematoxylin for blue nuclear stain. A representative image is shown (*n* > 10). (**C**) Quantification of Ki67 positive cells per image for different genotype groups of lung tumors (*n* > 25 image, * *p* < 0.05). (**D**) Western blot analysis for p-Akt and p-Erk of samples from normal lung tissues, SKT, and PKT tumors. Actin was used as loading control. The uncropped blots are shown in Appendix A.

**Figure 5 cancers-13-00974-f005:**
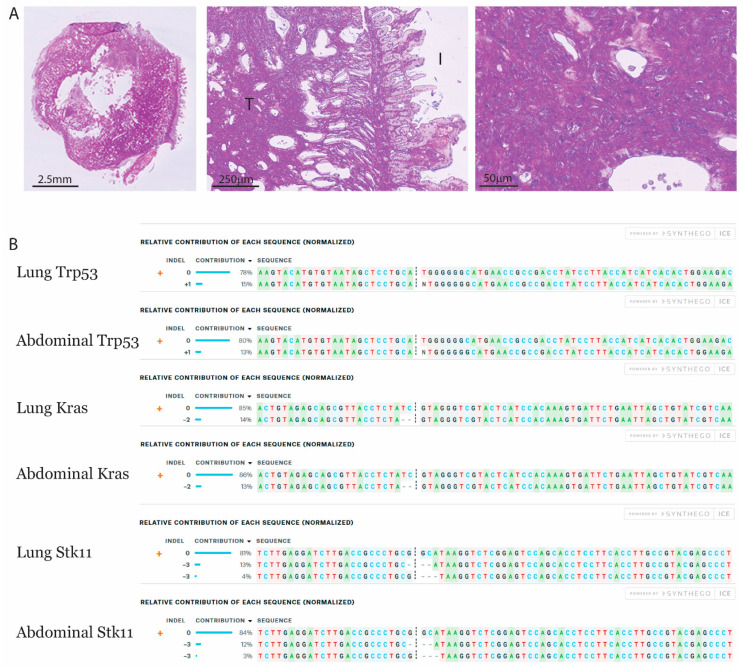
Abdominal metastasis from a primary lung tumor. An abdominal metastasis was present in a mouse with STK induced lung tumor 4 months post-viral transduction. (**A**) H&E stained section of a 5 mm metastasis in the abdomen. The metastasis was associated with the intestine, as intestinal villi were observed (I: intestine, T: tumor). (**B**) Sanger sequencing was performed on the metastasis and the primary lung tumor for target sites of the guide RNAs. The mutation profile was analyzed with ICE software and compared between the samples.

**Table 1 cancers-13-00974-t001:** Occurrence of STK11, PTEN, TP53, and KRAS mutations in lung cancer.

A	B	Neither	A Not B	B Not A	Both	Log2 Odds Rao	*p*-Value	*q*-Value	Tendency
**STK11**	**PTEN**	926	116	100	2	−2.647	<0.001	<0.001	Mutual exclusivity
**TP53**	**PTEN**	354	688	14	88	1.693	<0.001	<0.001	Co-occurrence
**KRAS**	**PTEN**	787	255	98	4	−2.989	<0.001	<0.001	Mutual exclusivity
**TP53**	**STK11**	300	726	68	50	−1.719	<0.001	<0.001	Mutual exclusivity
**STK11**	**KRAS**	830	55	196	63	2.278	<0.001	<0.001	Co-occurrence
**TP53**	**KRAS**	226	659	142	117	−1.823	<0.001	<0.001	Mutual exclusivity

## Data Availability

The data presented in this study are available in this article and Appendix A.

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
