# Peer review of "Comparative Analysis of Stk11/Lkb1 versus Pten Deficiency in Lung Adenocarcinoma Induced by CRISPR/Cas9"

_cancers, 2021, doi:10.3390/cancers13050974_

Round 1

Reviewer 1 Report

Berthelsen et al. here compared the effect of Stk11 and Pten on lung cancer, and provided experimental evidences that stk11 is essential for adenocarcinoma progression employing CRISPR/Cas9 technologies to model cancer pathophysiology. This study is well designed and performed logically, therefore I'm happy to read author's conclusions.

Author Response

We thank the reviewer for a positive evaluation of our work.

Reviewer 2 Report

In this manuscript by Berthelsen et.al., the authors have used the approach of using three different sgRNAs against Stk11, Pten and endogenous Kras expressed by adeno-associated viruses and replacement of Kras with oncogenic KRASG12D using the CRISPR system in LSL-Cas9 mice.  This allows clonal selection associated with tumor progression and hence supposed to better recapitulate human lung tumorigenesis compared to the traditional transgenic or knock-in approaches. Overall, this is a relevant model to study lung adenocarcinoma genesis by oncogenic KRASG12D and loss of Trp53 in the context of loss of key tumor suppressors, Stk11, or  Pten. The authors show that loss of Stk11 accelerates tumorigenesis compared to a loss of Pten. The specific concerns for this study are:

  1. The primary concern for this study is the fact that the authors have not been able to demonstrate the targeting and replacement of KRASG12D in most of the tumors; hence it is unclear what the oncogene is in the tumors that do not have KRASG12D by Sanger sequencing. For example, only 4/20 tumors in the STK group display KRASG12D repair, loss of Trp53, and Stk11. Is their approach of Sanger sequencing insensitive to detect the oncogenic KRAS because of contamination with normal lung cells?  Microdissection of tumors is one approach for sequencing. It would be nice to perform IHC analysis in parallel sections with the KRASG12D antibody (Abcam antibody ab221163 is one such antibody). Also very concerning is the lack of demonstration of oncogenic KRAS expression in the Pten deleted PKT tumors (Fig 4A) than the Stk11 deleted tumors. Since the primary finding here is the accelerated tumorigenesis in Stk11 deleted tumors, it is important only the tumors that have KRASG12D and loss of Trp53 compared for proliferation and size between loss of Stk11 or Pten.
  2. The authors describe these tumors as adenocarcinomas. It is hard to conclude from the histology shown from the small tumors. They should stain these sections with anti-Ttf1 to stain for type II cells and marker of adenocarcinoma histology. 
  3. Fig 3D- similar low power image showing both lungs as shown for Pten depleted tumors should be shown for the Stk11 tumors. The quality of these images is also poor.  It looks the overall tumor burden is more in Pten depleted tumors (although it seems true that Pten depleted tumors are smaller in size).
  4. The survival analysis in Fig. 3C should also show the controls of KRASG12D and Trp53 depleted mice. A composite figure of Fig 3C and Supplementary Fig. 3B will be better here.  Also, the number of mice should be mentioned in the figure itself. 
  5. The demonstration of the metastasis in Fig. 5 is convincing. However, the quality of the H&E images in Fig5A could be better. Also, here Ttf1 staining to demonstrate the tumor part will be helpful. 
  6. Significant editing of the manuscript will be needed. For example, Line 88 should be “was analyzed”; Line 218 in figure legend- “were examined” rather than “exanimated”.
  7. Line 235 in Discussion: the statement “The majority of the analyzed tumor samples had a mutation in all three target genes” is not true (Fig 4A). 
  8. Table 1: authors show mutual exclusivity of KRAS mutation and TP53 mutation. They should discuss why then they wanted to delete TP53 and express KRASG12D simultaneously and then compare the loss of either Stk11 or Pten.  What may happen if either Pten or Stk11 is deleted in the presence of only KRASG12D and the presence of Trp53. The authors should discuss this in Discussion if this particular experiment has not been conducted.   

Author Response

We thank the reviewer for through and constructive comments on this manuscript. We have addressed majority of the questions and comments within the available time. We hope the reviewer finds our answers appropriate. We are happy to elaborate further if required.  

  1. The primary concern for this study is the fact that the authors have not been able to demonstrate the targeting and replacement of KRASG12D in most of the tumors; hence it is unclear what the oncogene is in the tumors that do not have KRASG12D by Sanger sequencing. For example, only 4/20 tumors in the STK group display KRASG12D repair, loss of Trp53, and Stk11. Is their approach of Sanger sequencing insensitive to detect the oncogenic KRAS because of contamination with normal lung cells?  Microdissection of tumors is one approach for sequencing. It would be nice to perform IHC analysis in parallel sections with the KRASG12D antibody (Abcam antibody ab221163 is one such antibody). Also very concerning is the lack of demonstration of oncogenic KRAS expression in the Pten deleted PKT tumors (Fig 4A) than the Stk11 deleted tumors. Since the primary finding here is the accelerated tumorigenesis in Stk11 deleted tumors, it is important only the tumors that have KRASG12D and loss of Trp53 compared for proliferation and size between loss of Stk11 or Pten.

The homolog repaired of CRISPR breaks are inefficient and occur in ~2%. In this study we see efficiencies on ~20-25%. This shows that gain of function in Kras drives lung adenocarcinoma formation. It is possible that other gain of function mutations has taken place in Kras after mutation induced by CRISPR. This has been reported before but we have not addressed it at the DNA level. Instead, we have performed WB and IHC for p-Erk to reveal if the down-stream pathway is active (fig 4 B, D). We have now included samples from the PKT and the KT (Kras+Trp53) groups, stained with p-Erk antibody to complement the WB results. This is included as a new supplementary figure (figure S4). We have added this to the manuscript at line 197.

Furthermore, we assess the proliferation in KrasG12D + Trp53 tumors and added the results to Figure 4 C. This has been added to the manuscript on line 204-5, and in the corresponding figure text.    

The authors describe these tumors as adenocarcinomas. It is hard to conclude from the histology shown from the small tumors. They should stain these sections with anti-Ttf1 to stain for type II cells and marker of adenocarcinoma histology. 

All the tumor samples presented in this study have been assessed by a pulmonary pathologist, who classified them as adenocarcinomas. We did not observe any squamous cell carcinoma tumors in this study. Staining for Ttf1 could confirm the origin as adenocarcinoma but we have not been able to perform those experiments in this short time.

  1. Fig 3D- similar low power image showing both lungs as shown for Pten depleted tumors should be shown for the Stk11 tumors. The quality of these images is also poor.  It looks the overall tumor burden is more in Pten depleted tumors (although it seems true that Pten depleted tumors are smaller in size).

We have increased the resolution of the images to improve the quality. Both images are taken from mice that have been sacrificed up on reaching humane endpoint. Therefore, the tumor burden in the Pten deficient mice are overall large, as many small tumors have developed at this time point. In contrast, the tumors with loss of Stk11 are larger but fewer tumors are detected at this time point. This has been clarified on line 159-161. 

  1. The survival analysis in Fig. 3C should also show the controls of KRASG12D and Trp53 depleted mice. A composite figure of Fig 3C and Supplementary Fig. 3B will be better here.  Also, the number of mice should be mentioned in the figure itself. 

We have followed the suggestions from the reviewer and have combined parts of the two figures. This has made additional changes to the manuscript and figure text. See line 165-166 and figure text. 

  1. The demonstration of the metastasis in Fig. 5 is convincing. However, the quality of the H&E images in Fig5A could be better. Also, here Ttf1 staining to demonstrate the tumor part will be helpful. 

As in point three, we have tried to increase the quality of the images. The metastasis has also been evaluated by our pathologist, who confirmed its origin to be from the lung epithelium. As mentioned by the reviewer, the Sanger sequencing data from the primary tumour and the metastasis, is strong evidence that these tumors arise from the same clonal event.     

  1. Significant editing of the manuscript will be needed. For example, Line 88 should be “was analyzed”; Line 218 in figure legend- “were examined” rather than “exanimated”.

We appreciated the corrections and have changed the text.

  1. Line 235 in Discussion: the statement “The majority of the analyzed tumor samples had a mutation in all three target genes” is not true (Fig 4A). 

This is not true for the Pten deficient samples. We have modified the statement as many of the Stk11 tumors had mutations in Kras but not with the intended gain of function mutation. Changes have been added to line 237.

  1. Table 1: authors show mutual exclusivity of KRAS mutation and TP53 mutation. They should discuss why then they wanted to delete TP53 and express KRASG12D simultaneously and then compare the loss of either Stk11 or Pten.  What may happen if either Pten or Stk11 is deleted in the presence of only KRASG12D and the presence of Trp53. The authors should discuss this in Discussion if this particular experiment has not been conducted.   

It is true that KRAS and TP53 mutations are predicted as “mutual exclusivity”. However, analysis of the TCGA data set mentioned in table 1 shows that a subset of patients samples harbor both mutations (117 samples). Therefore, co-occurrence of KRAS and TP53 are present in human samples and cannot be seen as an absolute “mutual exclusivity” event.

The advantage with the CRISPR guide induced mutations is the heterogeneity that is created with clones with only one, two or three mutations. Therefore, it is likely that tumors with mutations in Pten or Stk11 have occurred together with a KrasG12D mutation. As an example, a tumor with Stk11 and KrasG12D mutation was identified (figure 4A). We have elaborated on the subject in the discussion on line 250-255.  

Reviewer 3 Report

Multiplexed genome editing with DNA endonucleases has wide application, including for cellular therapies, and disease modelling, as is the case of the present paper. Where the author successfully edited STK11, PTEN KRAS and TP53. Although the study was well conducted with appropriate analysis. From my point of view, the study has a potential  limitation,  attributed to a chromosomal translocation,  as a natural by products of inducing simultaneous genomic breaks. The authors should discard this potential effect of simultaneous genome editing.  A thorough translocation frequency analysis using  orthogonal methods (droplet digital PCR, unidirectional sequencing, and metaphase fluorescence in situ hybridization) should be considered, in order to discard that the observed phenotype is strictly associated with the triple mutants and not to “contaminating” translocation due to the undesired  chromosomal translocation,  

Author Response

Multiplexed genome editing with DNA endonucleases has wide application, including for cellular therapies, and disease modelling, as is the case of the present paper. Where the author successfully edited STK11, PTEN KRAS and TP53. Although the study was well conducted with appropriate analysis. From my point of view, the study has a potential  limitation,  attributed to a chromosomal translocation,  as a natural by products of inducing simultaneous genomic breaks. The authors should discard this potential effect of simultaneous genome editing.  A thorough translocation frequency analysis using  orthogonal methods (droplet digital PCR, unidirectional sequencing, and metaphase fluorescence in situ hybridization) should be considered, in order to discard that the observed phenotype is strictly associated with the triple mutants and not to “contaminating” translocation due to the undesired  chromosomal translocation,  

We thank the reviewer for constructive comments and suggestions on the manuscript. The CRISPR model has its limitations and advantages, as stochastic events can be introduced, which will not be seen in classical mouse models with use of the Cre-lox system. The advantage of the CRISPR method is generation of cells with different mutations profile. Here our molecular analysis has been limited to Sanger sequencing, WB and IHC. CRISPR induced mutations could have generated a large deletion or a chromosomal translocations, which we have not identified. Our Sanger sequencing is based on PCR products that are 400-800 bp long. If a large deletion or chromosomal translocations has been generated, a PCR product will not be present from the tumor clone and instead wiltype DNA will be amplified. Therefore, we have complemented the Sanger sequencing with WB and IHC analysis. Whole genomic sequencing would be very informative to address different genetic events that have occurred in the tumors but this is beyond the scope of this manuscript.

We have discussed the issues on line 239-243.  

Round 2

Reviewer 2 Report

The authors have made minor changes to the manuscript. One main issue still remains is they have not demonstrated the introduction of KRASG12D. The reason could be Sanger Sequencing.  They should discuss this as a limitation in discussion and clearly state that they have not been able to detect the KRASG12D by sequencing because of technology.  NGS (Illumina) sequencing may have been able to detect the mutation.  Hence for the whole study, we are relying on the surrogate of downstream signaling of pERK that can happen by other co-occurring genetic changes.

Other minor points: In Results in Line 116-117:  they write Three different AAV particles were injected.  What about the KT control?  They mention that later in describing the figure. The quality of Fig 5 in PDF is poor- the sequence is not legible.    

Author Response

Reviewer #2 (Comments to the Author):

The authors have made minor changes to the manuscript. One main issue still remains is they have not demonstrated the introduction of KRASG12D. The reason could be Sanger Sequencing.  They should discuss this as a limitation in discussion and clearly state that they have not been able to detect the KRASG12D by sequencing because of technology.  NGS (Illumina) sequencing may have been able to detect the mutation.  Hence for the whole study, we are relying on the surrogate of downstream signaling of pERK that can happen by other co-occurring genetic changes.

ANSWER: We thank the reviewer for the comments on this manuscript. The reviewer finds that we have not demonstrated KrasG12D mutation in the lung tumor samples. We have assessed all the tumor samples for KrasG12D mutation with use of Sanger Sequencing. This has been illustrated in figure 4a and commented on in line 186-189 and 243-247. To illustrate the Sanger Sequencing analysis we have included a new supplementary figure (Figure S4) showing a sample with CRISPR induced KrasG12D mutation. This figure will complement the graph in figure 4a and has been labelled in the text on line 189.  

Other minor points: In Results in Line 116-117:  they write Three different AAV particles were injected.  What about the KT control?  They mention that later in describing the figure. The quality of Fig 5 in PDF is poor- the sequence is not legible.  

ANSWER: We have made the correction to line 116-117 as we have been using four different AAVs. We thank the reviewer for the comment.

ANSWER: We have increased the resolution of the images in figure 5 to improve the quality. It is important that the sequence is legible.

Reviewer 3 Report

Thanks you for adressing all the points. 

Author Response

(The authors gave the same response as above.)
